# Experiences of mothers and health workers with MomCare and SafeCare bundles in Kenya and Tanzania: A qualitative evaluation

**Jonathan Izudi**[1,2]*, **Henry Odero Owoko**[1], **Moussa Bagayoko**[1], **Damazo Kadengye**[1,3]

**1** Data Science and Evaluation Unit, African Population and Health Research Center (APHRC), Nairobi, Kenya, **2** Department of Community Health, Faculty of Medicine, Mbarara University of Science and Technology, Mbarara, Uganda, **3** Department of Economics and Statistics, Kabale University, Kabale, Uganda

* jonahzd@gmail.com, jizudi@aphrc.org

**Data Availability Statement:** All relevant data are within the manuscript and its Supporting Information files.

## Abstract

Between 2019 and 2022, the digital dividend project (DDP), a technology-based intervention that combined care (MomCare) and quality improvement (SafeCare) bundles to empower mothers to access quality care during pregnancy, labor, and delivery, and postnatally, was implemented in Kenya and Tanzania aiming to improve maternal and newborn health outcomes. We describe the experiences of the mothers in accessing and utilizing health services under the bundles, and the experiences of the health workers in providing the services. Between November and December 2022, we conducted a qualitative evaluation across health facilities in Kenya and Tanzania. We held Interviews with mothers (pregnant and postpartum women who had benefited from the care bundles) and health workers (physicians, nurses, and midwives who provided the care bundles, including health facility In-Charges) at the antenatal care (ANC), skilled birth attendance (SBA), and postnatal care (PNC) service delivery points. We performed content analysis. Findings are reported using themes and quotes from the participants. We included 127 mothers (Kenya = 76, Tanzania = 51) and 119 health workers. Findings revealed that among mothers, the care bundles eased access to health services, ensured easy access and optimal ANC use, provision of respectful care, removed financial constraints, and led to the receipt of sufficient health education. Health workers reported that the care bundles offered them a new opportunity to provide quality maternal and newborn care and to adhere to the standard of care besides experiencing a positive and fulfilling practice. Health systems improvements included prompt emergency response and continual care, infrastructural developments, medical supplies and logistics, staffing, and increased documentation. Overall, the care bundles led to the strengthening of the healthcare system (staffing, service delivery, financing, supplies/logistics, and information management) in order to deliver quality maternal and child health services. The bundles should be replicated in settings with similar maternal and child health challenges.

**Funding:** The evaluation was supported by Children's Investment Fund (CIFF), Kenya. The funder had no role in study design, data collection and analysis, decision to publish, or preparation of the manuscript. There was no additional external funding received for this study.

**Competing interests:** The authors have declared that no competing interests exist.

## Introduction

Limited access to and non-use of maternal and child health services contribute to significant maternal and newborn morbidity and mortality. Globally, an estimated 810 women die daily from preventable causes related to pregnancy and childbirth, with 99% of the deaths being in sub-Saharan Africa (SSA) where access to quality maternal and child health services is limited [1]. Maternal deaths are a result of direct causes—excessive blood loss, infection, high blood pressure, unsafe abortion, and obstructed labor [1] and indirect causes—anemia, malaria, tuberculosis, human immunodeficiency virus (HIV), and heart diseases among others [2]. These causes are all preventable and treatable provided access to quality antenatal care (ANC), skilled birth attendance (SBA), and postnatal care (PNC) are guaranteed. The risk of under-five mortality in SSA is 15 times higher than in developed regions despite a declining global trend [3], with the leading causes as preterm delivery, pneumonia, birth asphyxia, diarrhea and malaria, and malnutrition among others. These causes are equally preventable and treatable provided access to quality child health services is assured.

Both Kenya and Tanzania have worse maternal and newborn outcomes. The 2022 Demographic and Health Survey data placed the maternal mortality ratio (MMR) in Kenya at 342 deaths per 100,000 live births, and in Tanzania at 524 deaths per 100,000 live births. To achieve the Sustainable Development Goal (SDG) target of reducing MMR to less than 70 deaths per 100,000 live births and neonatal mortality to below 12 deaths per 1,000 live births, context-specific interventions are urgently needed. One such intervention entails using digital technologies in health. For example, mobile phone technologies (sometimes called telehealth, telemedicine, e-health, or mhealth) have recently emerged as promising tools to improve access and use of maternal and child health services, with the ultimate goal being better maternal and child health outcomes [4]. The benefits of digital technologies include removing barriers to accessing maternal health services by increasing economic and geographic convenience [5]; promoting health in general but also increasing access to health education, health management, and health research; and, increasing access to health care [6], with the added benefits of increased knowledge as information about pregnancy and newborn health become readily available [7]. Other benefits include increasing the frequency of ANC visits, timing, and quality of services [8–10]. Telemedicine and phone-based referral networks are recommended solutions for tackling the decline in PNC service availability and utilization during and after the recent COVID-19 lockdown [11].

According to a recent systematic review and meta-analysis of data from low and lower-middle-income countries, mHealth increases SBA by more than twofold (pooled odds ratio [OR] 2.21, 95% CI 1.61–2.89), PNC use by nearly more than four-fold (pooled OR 4.13, 95% CI 1.90–8.97), and exclusive breastfeeding by nearly 2.3-fold (OR 2.25, 95% CI 1.46–3.46) [12]. Another systematic review and meta-analysis in low- and middle-income countries found that text messaging and mobile phone use improved breastfeeding within the first hour of life (pooled OR 2.01, 95% CI.27-2.75), as well as exclusive breastfeeding at 3–4 months (OR 1.88, 95% CI 1.26–2.50) and 6 months (OR 2.57, 95% CI 1.46–3.68) [13].

By supplementing in-person visits with mobile applications, mental health outcomes during pregnancy became comparable or even better, and the replacement of in-person visits with reduced prenatal care visits using telehealth for low-risk pregnancies led to similar clinical outcomes and higher patient satisfaction with care [14]. Recent systematic review and meta-analysis showed that remote breastfeeding support using digital technologies significantly reduced the risk of exclusive breastfeeding cessation at 3 months by 25% [15].

In 2019, the MMR in Kenya was 362 deaths per 100,000 live births [16], and that in Tanzania was 578 deaths per 100,000 live births [17]. The 5-year neonatal mortality rate (NNMR) in

Kenya was 22 per 1,000 live births and the infant mortality rate (IMR) was 39 per 1,000 live births [18]. In Tanzania, the NNMR was 25 deaths per 1,000 births and the IMR was 43 deaths per 1,000 births [19]. These rates are among the highest in SSA. One of the reasons for the poor maternal and child health outcomes is a lack of adequate access to quality health services in certain regions within and between the countries. In Kenya, the health facilities in Kisumu and Kakamega counties had poor maternal and child health outcomes while in Tanzania, it was the health facilities in the Manyara and Kilimanjaro regions. Prior to the digital intervention, in Kenya, the proportion of ANC attendance in Kakamega was 96.4% and that in Kisumu was 98.5%, with 47.0% SBA in Kakamega and 69.5% in Kisumu, and PNC checkup in the first two days after birth at 34.6% in the western region of Kakamega and 61.0% in the Nyanza region of Kisumu [20]. Tanzania's average four or more ANC visits was 51%, while it was rather high in the Manyara (98.2%) and Kilimanjaro (94.5%) regions. SBA was 47.5% in Manyara versus 95.5% in Kilimanjaro, but PNC checkup in the first two days after birth was significantly higher in Kilimanjaro than in Manyara (59.2% versus 26.5%, respectively) [21]. Therefore, a digital dividend project, a technology-based intervention that combined a care bundle (MomCare) and a quality improvement bundle (SafeCare), was started in both Kenya and Tanzania.

The aim was to increase access to quality maternal health services during pregnancy, labor and delivery, and postnatal periods, with the overall goal of improving maternal and newborn health outcomes. The project intended to achieve the outcomes through 1) contracting health workers to provide quality care for women; 2) assessing and improving the quality of care provided through SafeCare standards and tools; 3) monitoring health facilities for quality of care to ensure optimal pregnancy journey at the best cost; 4) providing women with the means to save or access subsidies including insurance and top-ups to pay for health services (the health wallet), and 5) rewarding health workers with bonuses in recognition for quality services that met all pre-agreed criteria for healthcare delivery. Despite the potential benefits of the care bundles, there is little known about the experiences of mothers and health workers with the bundles. Understanding these experiences is critical in order to inform the programming and implementation of the digital dividend project—recognizing the program failures and successes, determining the best approaches for addressing the failures, and taking full advantage of the successes in future programming. We, therefore, described the experiences of mothers in accessing and utilizing MomCare and SafeCare bundles during antenatal care, labor and delivery, and postnatal care. We also described the experiences of health workers in providing health services to mothers, both in Kenya and Tanzania between 2019 and 2022.

## Description of MomCare and SafeCare bundles

Between 2019 and September 2022, a three-year digital dividend project was implemented in Tanzania and Kenya by PharmAccess Foundation International, with funding from the Children's Investment Fund Foundation (CIFF). The project combined care (MomCare) and quality improvement (SafeCare) bundles to empower women to access the care they trust throughout their pregnancy journey and the postnatal period. Women were enrolled in a subsidized health insurance program and check-in and all the clinic costs were paid using a mobile platform. The service included "nudges" to remind women regarding check-ups and rewards to improve commitment. At the end of the care journey, health workers are compensated financially for providing high-quality care. Also, health workers were financially rewarded for positive health outcomes at the end of the care journey thus directly incentivizing quality and ensuring patient-centered care. The care journey for a mother begins at no more than 26 weeks of gestation except for teenagers and women living with human immunodeficiency

virus (HIV), who could be enrolled at any point during the journey. The enrolment ended at 20 weeks post-delivery, marking the 60[th] week of the care journey. Each step of the care journey was digitally tracked, and data were collected and analyzed to track and improve healthcare delivery.

SafeCare package is an internationally recognized standard-based quality improvement and recognition approach that has been operational since 2010 [22]. To improve the efficiency and cost-effectiveness of the SafeCare process, a digital assessment tool was developed. With this tool, SafeCare provides each health facility with a SafeCare quality score on a scale of 1–5 whenever a mother uses a service followed by a discussion of an automated quality improvement plan on the same day. In addition, the tool has automatic cross-checks allowing a reduced need and time for manual assessment and reviews by supervisors.

## Methods and materials

### Study design and setting

This qualitative evaluation was conducted in the two East African countries of Kenya and Tanzania between November and December 2022. In Kenya, 16 health facilities in Kisumu and Kakamega counties were included while in Tanzania, 51 health facilities from the Manyara and Kilimanjaro regions were included. The disparity in the number of health facilities analyzed could be attributed to current health needs, increased demand for health services due to population density, and the desire to expand healthcare coverage, among other considerations. In both countries, the regions selected had high maternal and neonatal morbidity and mortality based on the respective Demographic and Health Survey (DHS) data. In Kenya, the regions had the worst maternal and child health indicators, with MMR at 495 deaths per 100,000 live births and IMR at 40 deaths per 1000 live births in Kakamega and 50 deaths per 1000 live births in Kisumu. In Tanzania, the nomadic nature of the communities limited the use of existing services due to financial and physical constraints. In both settings, data were collected among participants at the ANC, SBA, and PNC service delivery points.

### Study population

We studied women aged 19–49 years who had benefited from MomCare and SafeCare bundles and health workers who provided care at the respective health facilities in Kenya and Tanzania. The women were purposively sampled based on the maximum variance sampling method, an approach that allowed equal representation of all women based on residence (rural versus urban), parity, insurance status, receipt of incentives, and level of education. The same sampling approach was applied to health workers (physicians, nurses, and midwives), health facility level, and years of experience (<5 versus ≥5 years).

At the managerial and leadership levels, we interviewed the health facility In-Charges, and for each of the categories of health workers, a minimum of five were included.

### Data collection

Data were collected through in-depth interviews (IDI) with mothers and health workers, and a focused group discussion (FGD) with mothers. Interviews were conducted in a noiseless place within the health facility premises, Monday to Friday, 9.00 am to 5.00 pm, lasting 30–45 minutes on average. The data collection focused on the experiences of mothers as beneficiaries of MomCare SafeCare bundles, and the health workers' experiences in providing maternal and child health services under the bundles. IDIs were conducted in the local language "*Kiswahili*" with the women and in the English language with health workers. FGDs were conducted by

two people, a moderator and a note-taker. Each group comprised 8–12 women with comparable age and parity.

## Quality control measures

Research assistants had ≥3 years of qualitative research experience and were drawn from the study area to ensure a better understanding of the local context and to ease the data collection. All research assistants trained in the health sciences discipline: nursing, clinical medicine, community health, and health nutrition. They received a 5-day training on the evaluation design, data collection process, and responsible conduct of research. Research assistants were organized into teams, each comprised of five people with one as the Team Leader. The Team Leader tracked the progress of the teams daily and provided technical assistance to the teams but with the support of the evaluation team. All data collection tools were pre-tested at a distant health facility not included in the evaluation. The feedback from pre-testing the tools was used to improve the implementation of the evaluation.

## Statistical issues

**Sample size estimation.**   No sample size was calculated but the number of people interviewed depended on *a priori sample size* deemed sufficient to achieve saturation, a point at which no new information emerges despite additional data collection as reported in previous qualitative studies [23–25]. Our *a priori sample size* estimate was 210 mothers (105 Kenya, 105 Tanzania) and approximately 150 health workers (75 Kenya, 75 Tanzania). We achieved saturation with 127 mothers (76 in Kenya, 51 in Tanzania) and 119 health workers.

**Data analysis.**   Data were collected through voice recordings and thereafter transcribed verbatim. The transcripts were verified by replaying the voice recordings. Any disparities between the transcript and the voice recordings were corrected. Field notes were scrutinized and compared with the audio recordings and the transcripts were cross-referenced with the field notes. Areas of departure were highlighted and discussed between the analysts. Transcriptions were done by 10 research assistants with experience in qualitative research. Dedoose software version 9.0.54 was used for the data analysis. The analysis was conducted by two independent female analysts to prevent subjective bias, and each analyst had ≥10 years of experience in qualitative research. The analysts coded the transcripts independently and developed the initial codes that were later harmonized through discussions and consensus to form the final codebook. The initial codes were then applied to the rest of the transcripts. The analysis adopted a thematic content approach and followed three steps, namely data immersion, coding, and coding sort. In data immersion, the two analysts (SJ and MN) familiarized themselves with the transcripts by reading and re-reading the transcripts several times to identify common and important texts and patterns. They allowed impressions to shape the data interpretations in different and unpredictable directions. SJ and MN then flagged the relevant parts of the transcripts with suitable words or codes and in the final stages, both analysts categorized the codes into themes and sub-themes in the agreed codebook. Three senior reviewers (MB, DK, and JI), with experience in qualitative and mixed-methods research, verified all emergent codes, themes, and sub-themes including the final codebook to minimize subject bias. Along with the participant's quotes, we presented the themes and sub-themes for the maternal and health worker experiences based on the World Health Organization's health systems strengthening building blocks as the guiding framework [26]. The six building blocks are service delivery, health workforce, financing, infrastructure and commodities, health information systems, and leadership and governance.

### Ethical consideration

We received ethical review and approval from the African Population and Health Research Center (APHRC) Internal Ethics Committee. The African Medical Research Foundation Ethical and Scientific Review Committee or AMREF-ESRC provided external review and ethical approval in Kenya (reference number: P911-2020). In Tanzania, the National Institute for Medical Research or NIMR (reference number: NIMR/HQ/R.8a/Vol.IX/3689) provided ethical clearance. All ethical approvals preceded the evaluation and all participants provided written informed consent.

Number tags and pseudo-names were used during data collection. Participation in the study was entirely voluntary and withdrawal from participation was permissible at any time.

## Results

### Characteristics of the participants

We summarize the participant's characteristics in Table 1. Overall, 127 mothers were included in the study, mainly from Kenya (n = 76) but not Tanzania (n = 51). The majority of the participants were from a rural setting, with parity ≥2, secondary or more levels of education, and at the PNC clinic. Health workers were mainly in the nursing and midwifery professions combined (n = 105).

### Main findings

We present the findings about the maternal and health worker experiences with the care bundles using the WHO's health systems building blocks as the guiding framework (Table 2).

### A. Maternal experiences

**Easy access and optimal ANC use.** Participants reported that MomCare improved access to ANC services. Many of the participants described the ANC service as being of good quality.

**Table 1. Participant characteristics and distributions.**

| Variables | Levels | Kenya (n = 76) | Tanzania (n = 51) | Overall (n = 127) |
|---|---|---|---|---|
| **Mothers** | | | | |
| Residence | Urban | 21 | 18 | 39 |
| | Rural | 44 | 29 | 73 |
| | Peri-urban | 11 | 4 | 15 |
| Parity | 1 | 26 | 10 | 36 |
| | ≥2 | 50 | 41 | 91 |
| Level of education | None | 2 | 1 | 3 |
| | Primary | 23 | 35 | 58 |
| | Secondary and over | 51 | 15 | 66 |
| Insurance status | Yes | 50 | 15 | 65 |
| | No | 26 | 36 | 62 |
| Point of service delivery | ANC | 22 | 25 | 47 |
| | SBA | 21 | 4 | 25 |
| | PNC | 33 | 22 | 55 |
| **Health workers** | **Levels** | | | |
| Type of health worker (n = 119) | Nurses and midwives | 62 | 43 | 105 |
| | Physician | 14 | 0 | 14 |

**Table 2. Summary of main themes and sub-themes according to the WHO health systems building blocks.**

| WHO health systems building blocks (themes) | A. Maternal experiences (sub-themes) | B. Health worker experiences (sub-themes) |
|---|---|---|
| 1. Service delivery | a) Easy access and optimal ANC use.<br>b) Respectful care provision.<br>c) Good quality care during childbirth<br>d) Sufficient health education and good care | a) New opportunity to provide quality care<br>b) Adherence to the standard of care<br>c) Positive and fulfilling practice<br>d) Better emergency response and continual care |
| 2. Health workforce | | Improved staffing |
| 3. Financing | No financial constraints to access care during the pregnancy journey | |
| 4. Infrastructure and commodities | a) Improved laboratory testing<br>b) Dispensing of all prescribed medications | Improved infrastructure, medical supplies, and logistics |
| 5. Health information systems | | Increased documentation |

We summarize the sub-themes based on five of the six domains of the WHO's health systems building blocks namely, service delivery, health workforce, financing, infrastructure and commodities, and health information systems. We did not find a sub-theme for the leadership and governance domain.

The participants were happy with the care bundle because it led to a successful pregnancy journey—the absence of complications during pregnancy.

'*What I can say about MomCare is that it is a nice program. It helps women who cannot even afford ANC, SBA, and PNC to have free services. There are some services MomCare also assists women with, sometimes they assist the women who require to undergo cesarean section (IDI, KM_42).*

'*MomCare is a good thing because there was a time I came when sick and got admitted like today and discharge the following day and I used it. Unlike others (insurance), you will be told that if you use it once, then you cannot use the card again for the second time that is why I like MomCare (IDI, KM_12).*

'*Upon reaching the 4th month, I used to come every week because the baby was not in a good position (IDI, TM_31).*

The participants reported having received their first ANC visit between the first and the fifth month of pregnancy, with the majority indicating to have started their first ANC visits in the fourth and third months of pregnancy. A few participants reported a late first ANC visit, which was either in the sixth or seventh month of pregnancy.

'*I came when my pregnancy was four months. I came because I was suspecting that I am pregnant. I was sick, and I was vomiting a lot. I could not eat anything or even take water. I was just vomiting, and the vomit resembled that of malaria. That is why I went to the hospital then I was told that I did not have malaria, I was pregnant. I was then told to start attending the clinic and I didn't waste time, the following week I started the clinic (IDI, KM_25).*

Participants indicated attending all scheduled ANC visits, with many reporting four to five ANC visits and a few reporting two to three ANC visits due to late first ANC attendance. Participants with high-risk pregnancies attended as many as 7–9 ANC as they needed close monitoring.

'*It (ANC visits) used to be every month depending on the date that you were scheduled. Let's say today is the 30th, you might be scheduled to come back next month on the 23rd. It*

*depends. Every month you go (ANC visit) depending on the return date that you have been given (IDI, KM_61).*

'*I came for the test, and they told me I am pregnant, then I went back and waited for four months before I started going to the clinic (meaning ANC clinic) many times, I think 5 to 6 times, yes. (IDI, TM_23).*

**Respectful care provision.** The majority of the women in Kenya and Tanzania indicate that the health workers treated them with respect and dignity during their ANC visits, a factor that motivated them to continue with all planned ANC schedules. For example, the health workers showed a positive attitude towards pregnant women during ANC visits, provided them with non-judgmental care, and ensured a conducive environment and open communication including respect—constituting respectful care. Overall, there was a good relationship between the health workers and mothers.

'*Me, what motivates me is the way I am being treated when I come for services. If you ask people about their experiences some will tell you they gave birth in a certain hospital, they were abused by nurses or if they are late, they were quarreled at so I also came with that fear of being quarreled at when I am late because I come from far.*

Sometimes, I fail to come. But, here, they are very gentle to us, and also, they have good services (FGD, Kenya, Mother).

'*What has motivated me in this facility is that they are just perfect. I have not been to a hospital like this, the place is clean, good services. They just serve you well, with respect. (IDI, TM_33).*

**Good quality care during childbirth.** It was stated that the quality of health services at the Momcare and SafeCare health facilities was good so many of the participants chose to receive both ANC and SBA services at the same health facility. The participants recommended the MomCare and SafeCare health facilities to other mothers because of the good quality care, friendly health workers during labor and delivery, easy access to health services, and affordable care.

'*I started here at Mukumu because I delivered all my children here. I have been coming to clinics here and also, even though I come from far, I prefer here because it has these services such as constant checking of the babies breathing, heart beating, the mother's condition, and even CS (cesarean section) if needed' (IDI, KM_07).*

'*When I arrived, the doctor put on gloves, he told me to lie on the bed, he amained my belly, after that, he was able to check if the cervix had dilated that is when he told me I must wait a bit, I had to do some exercise here. Together with one nurse we went around and came back.*

During the day she ensured I ate well, that night we slept here (at the facility) with the nurses, the second day they woke me up at five o'clock we went for some exercise, and at six o'clock they examined me, they told me that my cervix had dilated, they encouraged me to take some tea and on that morning at six o'clock I delivered' (IDI, TM_12).

**Sufficient health education and good care.** Complications during the PNC period are unpredictable both for the mother and the newborn so information on when to seek help is important. The participants indicated that the health workers provided them with sufficient health information during PNC visits to ensure their safety and that of the newborns.

'*After delivery, they continue to educate you on how to bathe the baby, then tell you to breast-feed the baby for six months*' *(FGD, Kenya).*

'*Immediately after delivery, they clean you and then they give you an injection to stop the bleeding and then you dress up and go to the resting bed. They observe you and if your status is okay, you are discharged the next day. You are told to go home. They give you a date to come for clinic and if the baby did not get the BCG vaccine and then there is the child's medical card. You follow up on that.*' *(IDI, TM_51)*

**No financial constraints during the pregnancy journey.** The participants mentioned that the bundles eased access to ANC, SBA, and PNC as the services became free due to health insurance schemes like the Linda Mama and the National Health Insurance Fund. In addition, they reported that the health insurance scheme provided them with comprehensive cover for their pregnancy journey like receiving more than one free ultrasound scan and undertaking several laboratory tests and other procedures.

'*I did not make any payments in the hospital. You know a hospital like Port Florence is private. When you go there, you will pay a lot of money but, when you have that card or NHIF (National Health Insurance Fund) you do not pay for anything*' *(IDI, KM_5).*

'*Free treatment. You would come to antenatal care for free, delivery was free and also after delivery, they were giving us small gifts for free*' *(IDI, KM_29).*

'*We used to pay for an ultrasound even if you go to a bigger hospital but now if you wanted an ultrasound you go to the clinic unit, they sign for you and get checked for free so we benefited in many things so I would like to request they should improve for us even more*' *(FGD, TM_52).*

**Improved laboratory testing.** Participants said the health workers performed several tests on them to check if they were at risk for complications during their pregnancy or delivery. Notable tests performed included those for HIV, COVID-19, syphilis, hepatitis, and high blood sugar levels among others.

'*Before going to the maternity ward, you have to get tested for COVID-19, HIV and. . . I was offered many tests and from there, they checked the position of the baby first because mine was done through elective CS (cesarean section) then, I was taken for surgery*' *(FGD, Kenya, Mother).*

'*MomCare brought all the tests now we can test mothers with all the required tests such as the HB (hemoglobin), blood level, syphilis, HIV, urinalysis, and blood group. When the mother comes for the first time, she has to get tested until when she is in labor. You monitor her during the clinic visits until she delivers. So, we have to do the tests that I told you earlier like HB (hemoglobin), urinalysis, VDRL (Venereal Disease Research Laboratory testing for syphilis), blood group, and ultrasound, although ultrasound is the last test.*' *(IDI, TH_24).*

**Dispensing of all prescribed medications.** A sufficient supply of medications or drugs is an important component of a strong health system. Through the MomCare and SafeCare bundles, all required medications became available during PNC visits. The participants indicated that they received all needed medications during their PNC visits.

"*All medications will be given to this mother as also immunization services we offer. In case the child is aged between 0–14 weeks we treat this child and also process referrals if we have*

*an emergency to refer the mother for further management, we just refer using our ambulance because it also helps the mother, it (MomCare) covers ambulance services.' (IDI, KM_13).*

"*The services we accessed during MomCare were much improved because we were well-considered in that, whenever the facility ran out of drugs, they were replenished on time. When I was not on the program (MomCare), I had to wait for the drugs to come from home by which time, I would have suffered because that took the time.' (FGD, Tanzania).*

## B. Healthcare provider experiences

### New opportunity to provide quality care

Health workers indicated that MomCare and SafeCare bundles presented them with a new opportunity to provide quality health services to mothers and their newborns. They felt motivated to provide all needed services as the bundles had incentives like training, regular support supervision, and follow-up by the implementing partners.

'*Basically, it (MomCare) has allowed us, the health workers, to provide quality health services to these women and babies through lab tests, ultrasound, and also medication like we have to give some medication in the process of labor and when they go home' (IDI, KH_68)*

'*It (meaning allowances) encourages a worker (meaning a health worker) who can do extra work, or extra hours to work. We have a few staff. So MomCare has truly helped to enlighten us.' (IDI, TH_44).*

### Adherence to the standard of care

Health workers indicated that the SafeCare bundles allowed them to adhere to the standards of care at all times. For instance, they indicated that the program permitted additional laboratory tests such as urinalysis, rhesus factor, and examinations like ultrasound scanning to be performed whenever needed.

'*I will manage the post-delivery process in case of a complication. I'm sure MomCare will take care of that. Then also the clients are free; the one that uses MomCare just comes in free. Even if she's sick, just comes in compared with the others' (IDI, KH_9).*

'*Because of MomCare, after a mother has delivered, she is now in PNC. We shall give her folate and vitamin K. You will also give her some eye ointment for the baby to prevent eye infections. You will observe her for 24 hours to see if there is going to be any challenge or not. After you confirm that the mother is in good health, you can release her to go home. You will give her appointments to come after 7 days and 21 days.' (IDI, TM_02).*

### Positive and fulfilling practice

The MomCare and SafeCare bundles were regarded as a blessing to health facilities and mothers. Health workers stated the program served women from all walks of life regardless of their socioeconomic status, residence, age, and HIV status.

In addition, they mentioned that by providing high-quality care to the mothers, they experienced a positive and fulfilling practice since all the mothers had positive outcomes at the end of their pregnancy journey.

'*That mother had four pregnancies and all of them were dying before delivery. The one we delivered now is the fourth. She had pressure, she had fibroids, and so on. So, the staff here began moving with her from day one of conception. So, when she reached six months, she was*

*more in danger because children used to die between five, six, and seven months. So, the doctors decided to operate. They delivered the baby at six months. We put it in our New Born Unit. Both survived and we are happy we helped. (IDI, KH_66).*

## Better emergency response and continual care

The majority of the health workers interviewed mentioned that the MomCare and SafeCare bundles led to a faster response to emergencies whenever needed thus reducing the risk of maternal deaths.

'*…………. they (mothers) do benefit in terms of emergencies. If you have a mother who is under MomCare and maybe she is in the village and goes into labor, and then she calls, we usually provide an ambulance. Do you see that as an advantage that others (mothers not covered by MomCare) would not get? (IDI, KH_11).*

'*There are big changes. MomCare has helped to reduce maternal deaths since it has supported those with low income, who could not afford some of the costs related to child delivery. Women come here without any cash, but the MomCare package caters to their needs. If there is any minor need for further medication, the facility usually top-ups. The same is done even in the case of surgery." (IDI, TH_20).*

'*When it comes to delivery, even at one time, it came a time when the program realized that there are women that time there was during COVID time (COVID-19), movements were restricted they provided ambulance services and women were ferried as long as they would call, the hospital would provide an ambulance and as long as is it is confirmed it is a MomCare mother they pay for it then, in maternity as they deliver, all those services and the expenses that she would incur the MomCare would cater for their payments (IDI, KH_17).*

## Improved staffing

Understaffing was a common problem at the health facilities before the MomCare and Safe-Care bundles were introduced. Many of the health workers interviewed stated that several MomCare nurses and midwives (health workers) were hired to keep pace with the ever-growing number of deliveries in the health facilities. According to the health workers, the hiring of new health workers under the MomCare and SafeCare bundles has helped to reduce their workload.

'*So, as a facility (health facility), I think we had less personnel by then, but the ones we have now are helping us run because we have hired five nurses, one clinical officer, and then of course, a lab tech (IDI, KH_08).*

## Increased documentation

Health workers noted that the project involved a lot of paperwork, especially during registration. This issue was worsened by the majority of the women being illiterate—increasing the workload for documentation among the health workers.

'*So, for negatives, I would only say that it (MomCare) had paperwork. It (MomCare) had a lot of paperwork that I did not like as a person and even the women because you see here, many people, don't even know how to write. They don't even know how to sign. So immediately they're treated, or they come for the ANC, there is a sheet that they were signing and*

*giving their phone numbers and other details and the women disliked that even though we were assisting, you cannot assist a person to put her signature (IDI, Kenya, HW_01).*

## Improved infrastructure, medical supplies, and logistics

Health workers reported that the MomCare and SafeCare guidelines helped to improve the health facility infrastructure as well as the availability of medical equipment and supplies. Such improvements reduced inter-health facility referrals, increased positive outcomes, and created a conducive workplace.

'*Facility (Health facility) has signage all over that will give you directions. You will know this is MCH (Maternal and Child Health Clinic), that is accounts, and that is administration, all courtesy of MomCare. The facility walls were also painted and the iron sheets were painted afresh. That was in 2019 it was courtesy of MomCare. The computers, most are courtesy of MomCare. Sometimes staff takes tea once in a while, courtesy of MomCare. A lot has happened. We have a new theatre, courtesy of MomCare. It's because of the increase in the number of clients (women) that the new theatre had to be built (IDI, KH_49).*

'*It (MomCare) has helped a lot by improving infrastructure that is offering delivery services, it has helped to build family planning facilities, and it has helped in buying maternity equipment and drugs. Those (health workers) who were offering services to the women also received allowances.' (IDI, TH_17).*

'*In the health facility, generally, our maternity has improved, and we have also improved our postnatal wards because these are the major areas, even the antenatal. We had to move from the other side and come to this side because of the number of clients and at least we have a space for all of them to be accommodated (IDI, TH_19).*

## Discussion

We report maternal experiences in accessing and using antenatal, skilled birth, and postnatal care, and the experiences of health workers in providing the services during the implementation of the MomCare and SafeCare bundles in Kenya and Tanzania. Before the implementation of the bundles, the majority of the mothers had shunned the health facilities as health services were of insufficient quality and access to health care was problematic due to high direct and indirect costs. In addition, the health workers were demotivated to provide care as the working environment was not conducive due to several factors: a lack of needed equipment, drug stockouts, and inadequate health staffing among others—constituting a weak health system.

Overall, we found the health system improved and became stronger during the implementation of the MomCare and SafeCare bundles. In particular, improvements are reported in maternal and neonatal health services delivery, staffing (human resources for health) supply of health commodities (sufficient drugs, supplies, and equipment), infrastructural development, and financing for maternal and neonatal health, all consistent with the World Health Organization's (WHO's) framework for health systems strengthening [27]. The WHO argues that a strong health system produces the desired quantity and quality of health services, has an adequate number of skilled health workers of an optimal mix, receives sufficient funding to support health service delivery, has a strong medical logistics and supply chain, and has a strong health management information system. With the implementation of the MomCare and SafeCare bundles, all six WHO's health systems strengthening blocks remarkably improved.

Our findings revealed positive experiences with the bundles among the mothers and health workers. Maternal experiences revealed easy access to health services, early and optimal ANC use, respectful care provision, absence of financial constraints, good quality care, receipt of all needed medications, and sufficient health education. Findings from health workers revealed a new opportunity to provide quality maternal and newborn care, adherence to the standard of care, positive and fulfilling practice, better emergency response and continual care, improved staffing, increased documentation at all three service delivery points, and infrastructure, medical supplies, and logistics.

Our findings of positive experiences among mothers and health workers are not surprising as several studies report digital technologies to improve the utilization of antenatal, skilled birth, and postnatal care. Digital health technologies are increasingly being used in several sub-Saharan African countries to improve maternal and child health services. Notable health services being improved using digital health technologies include ANC, SBA, and PNC among others [28]. One study conducted in Southern Tanzania showed that a digital health intervention improved neonatal healthcare outcomes, namely temperature control by keeping the neonates dry and warm, cord-cutting practices among health workers, and breastfeeding practices among mothers, including better preparation for obstetric care among expectant mothers through birth preparedness and complication readiness plans [28]. Our findings are consistent with the previous studies. Our findings are also in agreement with a previous study conducted in Tanzania that showed the use of digital health solutions for high-risk pregnancies improves the identification of women at risk for obstetric complications and the subsequent referral to higher-level health facilities [29].

The bundles led to positive experiences among health workers regarding service provision, which is not surprising. One previous study conducted about health workers' knowledge and attitudes towards the use of digital technologies in the provision of maternal health services at Tumbi Regional Referral Hospital in Tanzania found increased use of digital health technologies [30]. The study further showed that health workers understood the importance of digital health technologies in improving maternal health services besides reporting a positive attitude towards digital health technologies [30]. In another Tanzanian study, a mobile job aid was successfully used to support the counseling of women about contraception [31], which is consistent with our findings about the better quality of care during the postnatal care period due to the bundles.

Our findings agree with the increasing use of digital health technologies to eliminate barriers to accessing health services in sub-Saharan Africa [5].

## Study strengths and limitations

The study strengths include large sample size, data collection from all categories of participants, and all three service delivery points (ANC, SBA, and PNC), and using the WHO health systems strengthening blocks as a framework to report our findings of maternal and health worker experiences with the care bundles. Limitations include a lack of baseline qualitative and quantitative data for comparison of the present findings and the use of a qualitative evaluation that cannot demonstrate causation (cause-effect relationship). The lack of quantitative (numerical) data about service indicators both before and after the evaluation to provide a benchmark for comparison is another limitation worth considering. Lastly, it should be noted that bundles complemented the standard of care. Therefore, the findings should be cautiously interpreted so as not to give the impression as if all outcomes are singly attributed to the bundles.

## Conclusion and recommendations

The implementation of the MomCare and SafeCare bundles strengthened the health system for better maternal and child health services delivery. Maternal experiences regarding access

and utilization of ANC, SBA, and PNC were largely positive. Health worker experiences revealed satisfaction with the delivery of health services including a positive and fulfilling practice. We recommend the replication of the bundles in settings with similar maternal and child health challenges in SSA and beyond.

## Supporting information

**S1 File. Codebook.**
(DOCX)

## Acknowledgments

We are overwhelmingly indebted to the study participants for their time in providing invaluable information. We are grateful to all research assistants for their support.

## Author Contributions

**Conceptualization:** Jonathan Izudi, Henry Odero Owoko, Moussa Bagayoko, Damazo Kadengye.

**Data curation:** Jonathan Izudi, Henry Odero Owoko, Moussa Bagayoko, Damazo Kadengye.

**Formal analysis:** Jonathan Izudi, Henry Odero Owoko, Moussa Bagayoko, Damazo Kadengye.

**Funding acquisition:** Henry Odero Owoko, Moussa Bagayoko, Damazo Kadengye.

**Investigation:** Jonathan Izudi, Henry Odero Owoko, Moussa Bagayoko, Damazo Kadengye.

**Methodology:** Jonathan Izudi, Henry Odero Owoko, Moussa Bagayoko, Damazo Kadengye.

**Project administration:** Henry Odero Owoko, Moussa Bagayoko, Damazo Kadengye.

**Resources:** Henry Odero Owoko, Moussa Bagayoko, Damazo Kadengye.

**Software:** Jonathan Izudi, Moussa Bagayoko, Damazo Kadengye.

**Supervision:** Henry Odero Owoko, Moussa Bagayoko, Damazo Kadengye.

**Validation:** Jonathan Izudi, Moussa Bagayoko, Damazo Kadengye.

**Visualization:** Jonathan Izudi, Moussa Bagayoko, Damazo Kadengye.

**Writing – original draft:** Jonathan Izudi, Henry Odero Owoko, Moussa Bagayoko, Damazo Kadengye.

**Writing – review & editing:** Jonathan Izudi, Henry Odero Owoko, Moussa Bagayoko, Damazo Kadengye.

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
