## [Decision Letter · Decision Letter 0]

30 Aug 2023

PONE-D-23-14716Experiences of mothers and health workers with a Care Bundle in Kenya and Tanzania: a qualitative evaluationPLOS ONE

Dear Dr. Jonathan Izudi,

Thank you for submitting your manuscript to PLOS ONE. After careful consideration, we feel that it has merit but does not fully meet PLOS ONE’s publication criteria as it currently stands. Therefore, we invite you to submit a revised version of the manuscript that addresses the points raised during the review process.

We look forward to receiving your revised manuscript.

Kind regards,

Richard Kalisa, MD, PhD

Academic Editor

PLOS ONE

Journal Requirements:

“The evaluation was supported by Children's Investment Fund (CIFF), Kenya. The funder had no role in study design, data collection and analysis, decision to publish, or preparation of the manuscript.”

“We thank the Children Investment Fund Foundation (CIFF) and PharmAccess International for supporting this evaluation. We are overwhelmingly indebted to the study participants for their time in providing invaluable information. We are grateful to all research assistants for their support.”

“The evaluation was supported by Children's Investment Fund (CIFF), Kenya. The funder had no role in study design, data collection and analysis, decision to publish, or preparation of the manuscript.”

Additional Editor Comments (if provided):

Thank you for considering PLoS ONE for your manuscript "Experiences of mothers and health workers with a Care Bundle in Kenya and Tanzania: a qualitative evaluation". Peer review of your manuscript is now complete and, based on the reports, a major decision has been made your consideration on the shared comments on which final decision will be made. Please you can go ahead to address the reviewer’s comments and submit them within 10 working days. 

Reviewers' comments:

Reviewer's Responses to Questions

**Comments to the Author**

1. Is the manuscript technically sound, and do the data support the conclusions?

Reviewer #1: Partly

Reviewer #2: Partly

2. Has the statistical analysis been performed appropriately and rigorously? 

Reviewer #1: N/A

Reviewer #2: N/A

3. Have the authors made all data underlying the findings in their manuscript fully available?

Reviewer #1: Yes

Reviewer #2: Yes

4. Is the manuscript presented in an intelligible fashion and written in standard English?

Reviewer #1: Yes

Reviewer #2: No

5. Review Comments to the Author

Reviewer #1: Review comments on Experiences with mothers and health care givers with a care bundle in Kenya and Tanzania

The authors have made a good attempt at qualitatively evaluating a sort of mhealth-driven maternal and newborn health improvement project. The work is well written but will be significantly improved if the authors address the following concerns;

• Include the period of data collection in the abstract

• The Introduction is well written. Lines 82-89 could be improved by quantifying the many improvements alluded to.

• Since much of the problem seems to centre around access, the authors are encouraged to speak more to it by adequately describing what it used to be before the digital dividend project got underway

• “To date, little is known about the experiences of mothers and health workers with MomCare.”………why is this a problem? What is to be lost if this ‘problem’ remains unaddressed. The problem statement and study rationale are not well defined.

• “Besides, the regions had suboptimal use of ANC, SBA, and PNC services”……..the authors should try to quantify this. For instance, adequate ANC coverage (at least 4 visits) was X% in ABC County compared to Y% in DEF County over the same period of time.

• More than 3x as many health facilities were involved in Tanzania than in Kenya. Can the authors explain why this is so?

• I am thinking it is better to use the participant ID codes instead of just ‘mother, Kenya’. That brings out better the individuality in the responses. Also, include the average time of the interviews and the number of FGDs conducted

• Sentence 301-303 doesn’t come out well. It appears to be missing something. Look at it again.

• Under ‘Respectful Care’, 2 participants make mention of “good services”. Can the authors delve deeper into what constitutes “good services”?

• The ‘Discussion’ will be better presented if the authors gave a description of what the health system and service delivery context looked like before the roll-out of MomCare. What services were available or unavailable before MomCare? That way, the readers can better situate the benefits derived from MomCare.

• MomCare offered a subsidized payment on insurance premiums and it shot up utilization of services. Much as this is a qualitative work, the authors can still beef things up with quantitative hospital data regarding ANC attendance, Skilled birth delivery and PNC attendance in the three years before MomCare so that we can see the kind of transition MomCare has brought. In my country, we have had absolutely free maternal care and delivery services since 2007. We have made good progress in the proportion of pregnant women attending ANC at least once and at least 4x but we still don’t have a 100%. Let us not create the impression MomCare has led to a universal utilization of maternal and newborn services.

• Can the authors situate their findings (some if not all) in relevant theories underlying health seeking behaviour and motivation?

• I am missing key challenges in the project. Were there none?

Reviewer #2: Thank you for the opportunity to review the manuscript titled: Experiences of mothers and health workers with a Care Bundle in Kenya and Tanzania: a qualitative evaluation.

The title: You are evaluating an experience of women and health workers on two bundles; this should be well shown in the title. Mom Care and Safe Care

Abstract:

- The abstract is unstructured.

- Should give the brief description of participants (? pregnant women, lactating women, type of healthcare workers)

- Themes are not clear and are too many. Should try to summaries and make them clear.

- Conclusion not aligned with the findings.

Overall: The abstract needs revision to be clear and focused.

Background

- Needs to revise and make readable with easiness, as no full stop and no space between sentence and reference.

- Line 75: Recheck 524 deaths per 100,00…Need to be per 100,000.

- 76 revisit the sentence, you may need to omit the word “or”

- Line 97-101 is the repetition of paragraph line 73 with different references. You may need to combine this to be one paragraph.

Overall: Revise language and arrange the paragraphs to get better floor

Methods:

- Move the description of Mom Care to the background.

- Sample size: include reference to justify your method of sample size estimation.

- Line 207: Why specifically female analyst.

General comment

- Needs English revision

6. PLOS authors have the option to publish the peer review history of their article (what does this mean?). If published, this will include your full peer review and any attached files.

Reviewer #1: No

Reviewer #2: No

---

## [Author Response · Author response to Decision Letter 0]

9 Sep 2023

Editorial comments

Journal Requirements:

1. Please ensure that your manuscript meets PLOS ONE's style requirements, including those for file naming. The PLOS ONE style templates can be found at https://journals.plos.org/plosone/s/file?id=wjVg/PLOSOne_formatting_sample_main_body.pdf and https://journals.plos.org/plosone/s/file?id=ba62/PLOSOne_formatting_sample_title_authors_affiliations.pdf.

Response. We have adhered to the PLOS ONE’s style requirements.

2. Thank you for stating in your Funding Statement: “The evaluation was supported by Children's Investment Fund (CIFF), Kenya. The funder had no role in study design, data collection and analysis, decision to publish, or preparation of the manuscript.” Please provide an amended statement that declares *all* the funding or sources of support (whether external or internal to your organization) received during this study, as detailed online in our guide for authors at http://journals.plos.org/plosone/s/submit-now. Please also include the statement “There was no additional external funding received for this study.” in your updated Funding Statement.

Response. We have revised the Funding Statement to incorporate the needed changes. The revised Funding Statement included in the manuscript reads “The evaluation was supported by Children's Investment Fund (CIFF), Kenya. The funder had no role in study design, data collection and analysis, decision to publish, or preparation of the manuscript. There was no additional external funding received for this study.”

Response. The amended Funding Statement has been included in the cover letter and it reads: “The evaluation was supported by Children's Investment Fund (CIFF), Kenya. The funder had no role in study design, data collection and analysis, decision to publish, or preparation of the manuscript. There was no additional external funding received for this study.”

Response. We have provided a matching funding information and financial disclosure sections now. Both now reads “The evaluation was supported by Children's Investment Fund (CIFF), Kenya. The funder had no role in study design, data collection and analysis, decision to publish, or preparation of the manuscript. There was no additional external funding received for this study.”

4. Thank you for stating the following in the Acknowledgments Section of your manuscript: “We thank the Children Investment Fund Foundation (CIFF) and PharmAccess International for supporting this evaluation. We are overwhelmingly indebted to the study participants for their time in providing invaluable information. We are grateful to all research assistants for their support.” We note that you have provided additional information within the Acknowledgements Section that is not currently declared in your Funding Statement. Please note that funding information should not appear in the Acknowledgments section or other areas of your manuscript. We will only publish funding information present in the Funding Statement section of the online submission form.

Response. We have removed the funding information from the Acknowledgement Section. The remain text reads “We are overwhelmingly indebted to the study participants for their time in providing invaluable information. We are grateful to all research assistants for their support.”

2. Please remove any funding-related text from the manuscript and let us know how you would like to update your Funding Statement. Currently, your Funding Statement reads as follows: “The evaluation was supported by Children's Investment Fund (CIFF), Kenya. The funder had no role in study design, data collection and analysis, decision to publish, or preparation of the manuscript.”

Response. We have removed any funding-related text from the manuscript. Our updated funding statement should read as follows: “The evaluation was supported by Children's Investment Fund (CIFF), Kenya. The funder had no role in study design, data collection and analysis, decision to publish, or preparation of the manuscript. There was no additional external funding received for this study.”

Response. We have indicated the amended statements in the cover letter and it reads as follows: “The evaluation was supported by Children's Investment Fund (CIFF), Kenya. The funder had no role in study design, data collection and analysis, decision to publish, or preparation of the manuscript. There was no additional external funding received for this study.”

Additional Editor Comments (if provided):

1. Thank you for considering PLoS ONE for your manuscript "Experiences of mothers and health workers with a Care Bundle in Kenya and Tanzania: a qualitative evaluation". Peer review of your manuscript is now complete and, based on the reports, a major decision has been made your consideration on the shared comments on which final decision will be made. Please you can go ahead to address the reviewer’s comments and submit them within 10 working days. 

Response. We are grateful for considering our manuscript for review and the wonderful comments from the reviewers which have improved the quality of the manuscript.

Reviewer #1: 

Review comments on Experiences with mothers and health care givers with a care bundle in Kenya and Tanzania. The authors have made a good attempt at qualitatively evaluating a sort of mhealth-driven maternal and newborn health improvement project. The work is well written but will be significantly improved if the authors address the following concerns.

Response. Taking for taking time to review our manuscript. Your comments have greatly improved the quality of our manuscript. In the revised manuscript, we have strongly considered your comment #12 so we have re-arranged the findings based on the World Health Organization’s (WHO’s) health systems building blocks, namely service delivery, health work force, financing, infrastructure and commodities, heath information systems, and leadership and governance. Our findings address 5 of the 6 domains of the blocks. We hope you will find our revised manuscript more favorable.

1. Include the period of data collection in the abstract.

Response. We have added the data collection period and the new text reads “Between November and December 2022, we conducted a qualitative evaluation across health facilities in Kenya and Tanzania.”

2. The Introduction is well written. Lines 82-89 could be improved by quantifying the many improvements alluded to.

Response. We have re-checked the citations and noted that the importance of digital technologies was reported qualitatively so we found it hard to quantify it. We propose to maintain the description in qualitative terms. Second, we have revised the text on lines 82-89 for clarity so it reads: “The benefits of digital technologies include removing barriers to accessing maternal health services by increasing economic and geographic convenience; promoting health in general but also increasing access to health education, health management, and health research; and, increasing access to health care[6], with the added benefits of increased knowledge as information about pregnancy and newborn health become readily available. Other benefits include increasing the frequency of ANC visits, timing, and quality of services.”

3. Since much of the problem seems to centre around access, the authors are encouraged to speak more to it by adequately describing what it used to be before the digital dividend project got underway.

Response. Thank you for this valuable comment. Problematic access to quality maternal and child health services mostly manifested inform of poor health outcomes. For these reasons, we provided a description of the poor health outcomes before the project implementation thus justifying a need for an intervention. The text reads "In 2019, the MMR in Kenya was 362 deaths per 100,000 live births, and that in Tanzania was 578 deaths per 100,000 live births. The 5-year neonatal mortality rate (NNMR) in Kenya was 22 per 1,000 live births and the infant mortality rate (IMR) was 39 per 1,000 live births. In Tanzania, the NNMR was 25 deaths per 1,000 births and the IMR was 43 deaths per 1,000 births. These rates are among the highest in SSA". 

In addition, we added a new text to emphasize the problem of access to health services and it reads " One of the reasons for the poor maternal and child health outcomes is a lack of adequate access to quality health services in certain regions within and between the countries."

4. “To date, little is known about the experiences of mothers and health workers with MomCare.”………why is this a problem? What is to be lost if this ‘problem’ remains unaddressed. The problem statement and study rationale are not well defined.

Response. We have revised the sentence for clarity thus providing a rationale for the evaluation of the project. The revised sentence reads as follows: “Despite the potential benefits of the care bundles, there is little known about the experiences of mothers and health workers with the bundles. Understanding these experiences is critical in order to inform the programming and implementation of the digital dividend project—recognizing the program failures and successes, determining the best approaches for addressing the failures, and taking full advantage of the successes in future programming.”

5. “Besides, the regions had suboptimal use of ANC, SBA, and PNC services”……..the authors should try to quantify this. For instance, adequate ANC coverage (at least 4 visits) was X% in ABC County compared to Y% in DEF County over the same period of time.

Response. Thank you for this comment. We have deleted the above sentence because it is a repeat of previous data.

6. More than 3x as many health facilities were involved in Tanzania than in Kenya. Can the authors explain why this is so?

Response. We agree regarding the huge difference in the number of health facilities reported in the study between Kenya and Tanzania—51 vs. 16, respectively. These figures are accurate and are based on geographical and population differences between the 2 countries, with Tanzania having a wider geographical scope and a higher population compared to Kenya. To improve access to health services for its population, it is inevitable that Tanzania had more health facilities than Kenya. Third, from a programmatic viewpoint, the project was implemented at more health facilities in Tanzania than Kenya hence the observed difference.

7. I am thinking it is better to use the participant ID codes instead of just ‘mother, Kenya’. That brings out better the individuality in the responses. Also, include the average time of the interviews and the number of FGDs conducted.

Response. We have included the average time for the interviews using the text “Interviews were conducted in a noiseless place within the health facility premises, Monday to Friday, 9.00 am to 5.00 pm, lasting 30-45 minutes on average”.

Regarding the number of participants in each focused group discussion (FGD), we had included it our earlier submission and the text was: “FGDs were conducted by two people, a moderator and a note-taker. Each group comprised 8-12 women with comparable age and parity”. This text has been maintained in the revised submission. Regarding using participant codes, the idea was to anonymize the data. Based on the comment, we have now added the participants unique codes. 

8. Sentence 301-303 doesn’t come out well. It appears to be missing something. Look at it again.

Response. We agree that the sentence was not clear as some words were missed. We have revised the text for clarity and it reads “The participants mentioned that the bundles eased access to ANC, SBA, and PNC as the services became free due to health insurance schemes like the Linda Mama and the National Health Insurance Fund. In addition, they reported that the health insurance scheme provided them with comprehensive cover for their pregnancy journey like receiving more than one free ultrasound scan and undertaking several laboratory tests and other procedures.”

9. Under ‘Respectful Care’, 2 participants make mention of “good services”. Can the authors delve deeper into what constitutes “good services”?

Response. Thank you for this valuable comment. We reported quotes from 2 participants although several participants talked about respectful care. As stated in the quotes, the mothers were pleased with the health services they received because the health workers handled them with care, never harassed or abused them, and treated them with dignity and respect. We have revised the sentence to strengthen the point and to bring to context that several participants actually talked about respectful care. 

“The majority of the women in Kenya and Tanzania indicate that the health workers treated them with respect and dignity during their ANC visits, a factor that motivated them to continue with all planned ANC schedules. For example, the health workers showed a positive attitude towards pregnant women during ANC visits, provided them with non-judgmental care, and ensured a conducive environment and open communication including respect—constituting respectful care. Overall, there was a good relationship between the health workers and mothers.”

10. The ‘Discussion’ will be better presented if the authors gave a description of what the health system and service delivery context looked like before the roll-out of MomCare. What services were available or unavailable before MomCare? That way, the readers can better situate the benefits derived from MomCare.

Response. We agree with this comment. We have provided a brief description of health status and health service delivery to set the pace for contrasting with the discussion points. 

The brief text reads: “Before the implementation of the bundles, the majority of the mothers had shunned the health facilities as health services were of insufficient quality and access to health care was problematic due to high direct and indirect costs. In addition, the health workers were demotivated to provide care as the working environment was not conducive due to several factors: a lack of needed equipment, drug stockouts, and inadequate health staffing among others—constituting a weak health system.”

11. MomCare offered a subsidized payment on insurance premiums and it shot up utilization of services. Much as this is a qualitative work, the authors can still beef things up with quantitative hospital data regarding ANC attendance, Skilled birth delivery and PNC attendance in the three years before MomCare so that we can see the kind of transition MomCare has brought. In my country, we have had absolutely free maternal care and delivery services since 2007. We have made good progress in the proportion of pregnant women attending ANC at least once and at least 4x but we still don’t have a 100%. Let us not create the impression MomCare has led to a universal utilization of maternal and newborn services.

Response. We agree to not overstating the benefits of MomCare in improving service utilization as it complemented the standard of care. Therefore, we have revised certain sections of the texts. We do not have quantitative data about maternal and child health outcomes as well as the service utilization, both of which have been stated as limitations to the study. The revised limitations section read as follows: “Limitations include a lack of baseline qualitative and quantitative data for comparison of the present findings and the use of a qualitative evaluation that cannot demonstrate causation (cause-effect relationship). The lack of quantitative data about service indicators both before and after the evaluation to provide a benchmark for comparison is another limitation worth considering. Lastly, it should be noted that bundles complemented the standard of care. Therefore, the findings should be cautiously interpreted so as not to give the impression as if all outcomes are singly attributed to the bundles.”

12. Can the authors situate their findings (some if not all) in relevant theories underlying health seeking behaviour and motivation?

Response. We appreciate the use of theories, models, or frameworks in presenting the findings of qualitative studies because it improves the rigor. Accordingly, we have adapted the World Health Organization’s (WHO’s) health systems strengthening building blocks framework in reporting the findings. We have added a text in the methods section, under the data analysis to indicate this change. The text reads “Along with the participant’s quotes, we presented the themes and sub-themes for the maternal and health worker experiences based on the World Health Organization’s health systems strengthening building blocks as the guiding framework. The six building blocks are service delivery, health work force, financing, infrastructure and commodities, heath information systems, and leadership and governance.”

13. I am missing key challenges in the project. Were there none?

Response. The project indeed had challenges. However, we felt that the challenges fell outside the scope of the question addressed by the current manuscript. Therefore, we did not report them.

Reviewer #2:

Thank you for the opportunity to review the manuscript titled: Experiences of mothers and health workers with a Care Bundle in Kenya and Tanzania: a qualitative evaluation.

Response. We thank you for dedicating time to rigorously read and review our manuscript, and to provide insightful comments which have now improved the quality. We are forever grateful. Below, we addressed each specific comment and where we had challenges in addressing any, we offered and explanation.

The title: 

1. You are evaluating an experience of women and health workers on two bundles; this should be well shown in the title. Mom Care and Safe Care

Response. We have revised the title and it reads: “Experiences of mothers and health workers with MomCare and SafeCare Bundles in Kenya and Tanzania: a qualitative evaluation”

Abstract:

2. The abstract is unstructured.

Response. The PLoS ONE requirement is to have an unstructured abstract hence the current abstract. We are happy to submit a structured abstract if required.

3. Should give the brief description of participants (? pregnant women, lactating women, type of healthcare workers)

Response. We have added a description of the participants. The new text reads: “We held Interviews with mothers (pregnant and postpartum women who had benefited from the care bundles) and health workers (physicians, nurses, and midwives who provided the care bundles, including health facility In-Charges) at the antenatal care (ANC), skilled birth attendance (SBA), and postnatal care (PNC) service delivery points.”

4. Themes are not clear and are too many. Should try to summaries and make them clear.

Response. We appreciate this comment and have revised the themes for clarity. Regarding the number of themes, we found it a little hard to reduce them given the objectives of the study. The themes and sub-themes are a summary of the important ones. Please note that reviewer #1 suggested that we used a model, or theory, or framework to situate our findings (comment #12). In response, we have used the WHO health systems building blocks as a framework to report the findings. These now reduced the themes to 5 but the sub-themes remain. We propose to maintain them.

5. Conclusion not aligned with the findings.

Response. Overall, the bundles led to a strengthened heath system for better delivery of maternal and child health services and our conclusion has been aligned to speak to this finding. Therefore, the text for the conclusion reads as follows: “Overall, the care bundles led to the strengthening of the healthcare system (staffing, service delivery, financing, supplies/logistics, and information management) in order to deliver quality maternal and child health services. The bundles should be replicated in settings with similar maternal and child health challenges.”

6. Overall: The abstract needs revision to be clear and focused.

Response. We have revised the abstract for clarity.

Background

7. Needs to revise and make readable with easiness, as no full stop and no space between sentence and reference.

Response. We have revised the background section, addressing all punctuation errors and other identifiable issues. 

8. Line 75: Recheck 524 deaths per 100,00…Need to be per 100,000.

Response. This has been corrected to read as “524 deaths per 100,000 live births.”

9. 76 revisit the sentence, you may need to omit the word “or”

Response. This has been corrected. The revised text reads “To achieve the Sustainable Development Goal (SDG) target of reducing MMR to less than 70 deaths per 100,000 live births and neonatal mortality to below 12 deaths per 1,000 live births, context-specific interventions are urgently needed.”

10. Line 97-101 is the repetition of paragraph line 73 with different references. You may need to combine this to be one paragraph.

Response. Thank you for this comment. The initial lines reported the most recent data about maternal mortality ratio (MMR) using the 2022 data. The text is " The 2022 Demographic and Health Survey data placed the maternal mortality ratio (MMR) in Kenya at 342 deaths per 100,000 live births, and in Tanzania at 524 deaths per 100,000 live births." The other lines, reported MMR in 2019—the data that motivated a need for the project. The text is "In 2019, the MMR in Kenya was 362 deaths per 100,000 live births, and that in Tanzania was 578 deaths per 100,000 live births." Both sections are thus different.

11. Overall: Revise language and arrange the paragraphs to get better floor.

Response. We have revised the manuscript for clarity and performed spelling and grammatically checks.

Methods:

12. Move the description of Mom Care to the background.

Response. We have added the description of MomCare to the background as recommended.

13. Sample size: include reference to justify your method of sample size estimation.

Response. We have provided reference to justify the principle of saturation as used in qualitative studies. The revised text reads as follows: “No sample size was calculated but the number of people interviewed depended on a priori sample size deemed sufficient to achieve saturation, a point at which no new information emerges despite additional data collection as reported in previous qualitative studies”. Reference numbers are #19-21.

14. Line 207: Why specifically female analyst.

Response. We do not have a particular reason for using female analysts. It was by chance that female analysts were recruited. Overall, no specific reason.

General comment

15. Needs English revision.

Response. We have read the manuscript and made needed revisions. We have done a spelling and grammatically checks. We believe the manuscript is clear now.

---

## [Decision Letter · Decision Letter 1]

30 Oct 2023

PONE-D-23-14716R1Experiences of mothers and health workers with MomCare and SafeCare Bundles in Kenya and Tanzania: a qualitative evaluationPLOS ONE

Dear Dr. Izudi,

Thank you for submitting your manuscript to PLOS ONE. After careful consideration, we feel that it has merit but does not fully meet PLOS ONE’s publication criteria as it currently stands. Therefore, we invite you to submit a revised version of the manuscript that addresses the points raised during the review process. Please submit your revised manuscript by Dec 14 2023 11:59PM. If you will need more time than this to complete your revisions, please reply to this message or contact the journal office at plosone@plos.org. Please include the following items when submitting your revised manuscript:A rebuttal letter that responds to each point raised by the academic editor and reviewer(s). You should upload this letter as a separate file labeled 'Response to Reviewers'.A marked-up copy of your manuscript that highlights changes made to the original version. You should upload this as a separate file labeled 'Revised Manuscript with Track Changes'.An unmarked version of your revised paper without tracked changes. You should upload this as a separate file labeled 'Manuscript'.If applicable, we recommend that you deposit your laboratory protocols in protocols.io to enhance the reproducibility of your results. Protocols.io assigns your protocol its own identifier (DOI) so that it can be cited independently in the future. For instructions see: https://journals.plos.org/plosone/s/submission-guidelines#loc-laboratory-protocols. Additionally, PLOS ONE offers an option for publishing peer-reviewed Lab Protocol articles, which describe protocols hosted on protocols.io. Read more information on sharing protocols at https://plos.org/protocols?utm_medium=editorial-email&utm_source=authorletters&utm_campaign=protocols.

We look forward to receiving your revised manuscript.

Kind regards,

Azmeraw Ambachew Kebede, MSc

Academic Editor

PLOS ONE

Journal Requirements:

Reviewers' comments:

Reviewer's Responses to Questions

**Comments to the Author**

1. If the authors have adequately addressed your comments raised in a previous round of review and you feel that this manuscript is now acceptable for publication, you may indicate that here to bypass the “Comments to the Author” section, enter your conflict of interest statement in the “Confidential to Editor” section, and submit your "Accept" recommendation.

Reviewer #1: (No Response)

Reviewer #2: All comments have been addressed

2. Is the manuscript technically sound, and do the data support the conclusions?

Reviewer #1: Yes

Reviewer #2: Yes

3. Has the statistical analysis been performed appropriately and rigorously? 

Reviewer #1: N/A

Reviewer #2: N/A

4. Have the authors made all data underlying the findings in their manuscript fully available?

Reviewer #1: Yes

Reviewer #2: Yes

5. Is the manuscript presented in an intelligible fashion and written in standard English?

Reviewer #1: Yes

Reviewer #2: Yes

6. Review Comments to the Author

Reviewer #1: Experiences of mothers and health workers with MomCare and SafeCare Bundles in Kenya and Tanzania: a qualitative evaluation. PONE-D-23-14716R1

General Comments

I thank the authors for incorporating the review comments and suggestions. On the bases of their responses, I still have a few comments for them to address. I therefore recommend a minor revision at this stage

1.Response. We have re-checked the citations and noted that the importance of digital technologies was reported qualitatively so we found it hard to quantify it. We propose to maintain the description in qualitative terms. Second, we have revised the text on lines 82-89 for clarity so it reads: “The benefits of digital technologies include removing barriers to accessing maternal health services by increasing economic and geographic convenience; promoting health in general but also increasing access to health education, health management, and health research; and, increasing access to health care[6], with the added benefits of increased knowledge as information about pregnancy and newborn health become readily available. Other benefits include increasing the frequency of ANC visits, timing, and quality of services.”

Regarding the above, I am sure you can find systematic reviews that will enable you quantify maternal and newborn health outcomes following trials with mHealth/digital technologies.

2.Response. Thank you for this valuable comment. Problematic access to quality maternal and child health services mostly manifested inform of poor health outcomes. For these reasons, we provided a description of the poor health outcomes before the project implementation thus justifying a need for an intervention. The text reads "In 2019, the MMR in Kenya was 362 deaths per 100,000 live births, and that in Tanzania was 578 deaths per 100,000 live births. The 5-year neonatal mortality rate (NNMR) in Kenya was 22 per 1,000 live births and the infant mortality rate (IMR) was 39 per 1,000 live births. In Tanzania, the NNMR was 25 deaths per 1,000 births and the IMR was 43 deaths per 1,000 births. These rates are among the highest in SSA". In addition, we added a new text to emphasize the problem of access to health services and it reads " One of the reasons for the poor maternal and child health outcomes is a lack of adequate access to quality health services in certain regions within and between the countries."

Outcomes such as those described in your manuscript reflect not only access but a myriad of other factors. I think your work would be clearer if you could quantify ‘access’ in terms of, for example, ANC attendance in XXX area was 54% or skilled delivery at YYYY hospital was 48%, etc before the digital intervention.……if you can find the data to support it

3.Response. We agree regarding the huge difference in the number of health facilities reported in the study between Kenya and Tanzania—51 vs. 16, respectively. These figures are accurate and are based on geographical and population differences between the 2 countries, with Tanzania having a wider geographical scope and a higher population compared to Kenya. To improve access to health services for its population, it is inevitable that Tanzania had more health facilities than Kenya. Third, from a programmatic viewpoint, the project was implemented at more health facilities in Tanzania than Kenya hence the observed difference.

Understood……but find a way to work this explanation into the manuscript…albeit brief.

4.The brief text reads: “Before the implementation of the bundles, the majority of the mothers had shunned the health facilities as health services were of insufficient quality and access to health care was problematic due to high direct and indirect costs. In addition, the health workers were demotivated to provide care as the working environment was not conducive due to several factors: a lack of needed equipment, drug stockouts, and inadequate health staffing among others—constituting a weak health system.”

Great! This reinforces my call for quantifying access in (2) abov

Reviewer #2: Thank you for responding to all comments I have raised.

The manuscript reads better and is focused.

The abstract and whole manuscript reads well.

7. PLOS authors have the option to publish the peer review history of their article (what does this mean?). If published, this will include your full peer review and any attached files.

Reviewer #1: No

Reviewer #2: No

---

## [Author Response · Author response to Decision Letter 1]

31 Oct 2023

Response to Reviewers

General Comments

I thank the authors for incorporating the review comments and suggestions. On the bases of their responses, I still have a few comments for them to address. I therefore recommend a minor revision at this stage.

Response. We thank the reviewers for taking time to rigorously review our manuscript. We received very valuable comments that have tremendously improved the quality of the manuscript.

1. Response. We have re-checked the citations and noted that the importance of digital technologies was reported qualitatively so we found it hard to quantify it. We propose to maintain the description in qualitative terms. Second, we have revised the text on lines 82-89 for clarity so it reads: “The benefits of digital technologies include removing barriers to accessing maternal health services by increasing economic and geographic convenience; promoting health in general but also increasing access to health education, health management, and health research; and, increasing access to health care[6], with the added benefits of increased knowledge as information about pregnancy and newborn health become readily available. Other benefits include increasing the frequency of ANC visits, timing, and quality of services.” 

Regarding the above, I am sure you can find systematic reviews that will enable you quantify maternal and newborn health outcomes following trials with mHealth/digital technologies.

Response. Thank you for the guidance. We have now added findings from systematic reviews and meta-analyses to the manuscript to quantify the effect of mHealth. The new text reads as follows: 

“According to a recent systematic review and meta-analysis of data from low and lower-middle-income countries, mHealth increases SBA by more than twofold (pooled odds ratio [OR] 2.21, 95% CI 1.61-2.89), PNC use by nearly more than four-fold (pooled OR 4.13, 95% CI 1.90-8.97), and exclusive breastfeeding by nearly 2.3-fold (OR 2.25, 95% CI 1.46-3.46). Another systematic review and meta-analysis in low- and middle-income countries found that text messaging and mobile phone use improved breastfeeding within the first hour of life (pooled OR 2.01, 95% CI.27-2.75), as well as exclusive breastfeeding at 3-4 months (OR 1.88, 95% CI 1.26-2.50) and 6 months (OR 2.57, 95% CI 1.46-3.68).”

2. Response. Thank you for this valuable comment. Problematic access to quality maternal and child health services mostly manifested inform of poor health outcomes. For these reasons, we provided a description of the poor health outcomes before the project implementation thus justifying a need for an intervention. The text reads "In 2019, the MMR in Kenya was 362 deaths per 100,000 live births, and that in Tanzania was 578 deaths per 100,000 live births. The 5-year neonatal mortality rate (NNMR) in Kenya was 22 per 1,000 live births and the infant mortality rate (IMR) was 39 per 1,000 live births. In Tanzania, the NNMR was 25 deaths per 1,000 births and the IMR was 43 deaths per 1,000 births. These rates are among the highest in SSA". In addition, we added a new text to emphasize the problem of access to health services and it reads " One of the reasons for the poor maternal and child health outcomes is a lack of adequate access to quality health services in certain regions within and between the countries."

Outcomes such as those described in your manuscript reflect not only access but a myriad of other factors. I think your work would be clearer if you could quantify ‘access’ in terms of, for example, ANC attendance in XXX area was 54% or skilled delivery at YYYY hospital was 48%, etc before the digital intervention.……if you can find the data to support it.

Response. Thank you for the valuable comment. We have quantified the access in terms of ANC, SBA, and PNC before the digital intervention and the new text reads as follows: 

“Prior to the digital intervention, in Kenya, the proportion of ANC attendance in Kakamega was 96.4% and that in Kisumu was 98.5%, with 47.0% SBA in Kakamega and 69.5% in Kisumu, and PNC checkup in the first two days after birth at 34.6% in the western region of Kakamega and 61.0% in the Nyanza region of Kisumu. Tanzania's average four or more ANC visits was 51%, while it was rather high in the Manyara (98.2%) and Kilimanjaro (94.5%) regions. SBA was 47.5% in Manyara versus 95.5% in Kilimanjaro, but PNC checkup in the first two days after birth was significantly higher in Kilimanjaro than in Manyara (59.2% versus 26.5%, respectively).”

3. Response. We agree regarding the huge difference in the number of health facilities reported in the study between Kenya and Tanzania—51 vs. 16, respectively. These figures are accurate and are based on geographical and population differences between the 2 countries, with Tanzania having a wider geographical scope and a higher population compared to Kenya. To improve access to health services for its population, it is inevitable that Tanzania had more health facilities than Kenya. Third, from a programmatic viewpoint, the project was implemented at more health facilities in Tanzania than Kenya hence the observed difference.

Understood……but find a way to work this explanation into the manuscript…albeit brief.

Response. Again, thank you the comment. We have inserted a brief explanation and it reads “The disparity in the number of health facilities analyzed could be attributed to current health needs, increased demand for health services due to population density, and the desire to expand health-care coverage, among other considerations.” 

4. The brief text reads: “Before the implementation of the bundles, the majority of the mothers had shunned the health facilities as health services were of insufficient quality and access to health care was problematic due to high direct and indirect costs. In addition, the health workers were demotivated to provide care as the working environment was not conducive due to several factors: a lack of needed equipment, drug stockouts, and inadequate health staffing among others—constituting a weak health system.”

Great! This reinforces my call for quantifying access in (2) above.

Response. Thank you for the compliments. We have added the required data about service utilization in comment #2.

New citations regarding the added texts:

1. Gayesa RT, Ngai FW, Xie YJ. The effects of mHealth interventions on improving institutional delivery and uptake of postnatal care services in low-and lower-middle-income countries: a systematic review and meta-analysis. BMC health services research. 2023;23(1):611. Epub 2023/06/10. doi: 10.1186/s12913-023-09581-7. PubMed PMID: 37296420; PubMed Central PMCID: PMCPMC10257264.

2. Lee SH, Nurmatov UB, Nwaru BI, Mukherjee M, Grant L, Pagliari C. Effectiveness of mHealth interventions for maternal, newborn and child health in low- and middle-income countries: Systematic review and meta-analysis. Journal of global health. 2016;6(1):010401. Epub 2015/12/10. doi: 10.7189/jogh.06.010401. PubMed PMID: 26649177; PubMed Central PMCID: PMCPMC4643860.

3. Kenya National Bureau of Statistics (KNBS), ICF. 2014 Kenya Demographic and Health Survey (KDHS). Nairobi, Kenya: KNBS and ICF., 2014.

4. Ministry of Health. Community Development G, Elderly, and Children (MoHCDGEC) [Tanzania Mainland],, Ministry of Health (MoH) [Zanzibar], National Bureau of Statistics, Office of the Chief Government Statistician (OCGS), ICF. Tanzania Demographic and Health Survey and Malria Indicator Survey (TDHS-MIS) 2015-16. Dare es Salaam: MoHCDGEC, MoH, NBS, OCGS, and ICF, 2016.

---

## [Editor Report · Decision Letter 2]

3 Nov 2023

Experiences of mothers and health workers with MomCare and SafeCare Bundles in Kenya and Tanzania: a qualitative evaluation

PONE-D-23-14716R2

Dear Dr. Izudi,

We’re pleased to inform you that your manuscript has been judged scientifically suitable for publication and will be formally accepted for publication once it meets all outstanding technical requirements.

Kind regards,

Azmeraw Ambachew Kebede, MSc

Academic Editor

PLOS ONE
---

## [Editor Report · Acceptance letter]

7 Nov 2023

PONE-D-23-14716R2 

Experiences of mothers and health workers with MomCare and SafeCare Bundles in Kenya and Tanzania: a qualitative evaluation 

Dear Dr. Izudi:

I'm pleased to inform you that your manuscript has been deemed suitable for publication in PLOS ONE. Congratulations! Your manuscript is now with our production department. 

Kind regards, 

on behalf of

Mr. Azmeraw Ambachew Kebede 

Academic Editor

PLOS ONE